# Home-based exercise for people living with frailty and chronic kidney disease: A mixed-methods pilot randomised controlled trial

Andrew C. Nixon[1,2,3], Theodoros M. Bampouras[4,5]*, Helen J. Gooch[3,6], Hannah M. L. Young[7,8], Kenneth W. Finlayson[9], Neil Pendleton[10], Sandip Mitra[11,12,13], Mark E. Brady[1], Ajay P. Dhaygude[1]

1 Department of Renal Medicine, Lancashire Teaching Hospitals NHS Foundation Trust, Preston, United Kingdom, 2 Division of Cardiovascular Sciences, University of Manchester, Manchester, United Kingdom, 3 Centre for Health Research and Innovation, National Institute of Health Research Lancashire Clinical Research Facility, Lancashire Teaching Hospitals NHS Foundation Trust, Preston, United Kingdom, 4 Lancaster Medical School, Lancaster University, Lancaster, United Kingdom, 5 The Centre for Ageing Research, Lancaster University, Lancaster, United Kingdom, 6 Core Therapies Department, Lancashire Teaching Hospitals NHS Foundation Trust, Preston, United Kingdom, 7 Department of Respiratory Sciences, University of Leicester, Leicester, United Kingdom, 8 Department of Research and Innovation, University Hospitals of Leicester NHS Trust, Leicester, United Kingdom, 9 Research in Childbirth and Health Unit, University of Central Lancashire, Preston, United Kingdom, 10 Division of Neuroscience and Experimental Psychology, University of Manchester, Manchester, United Kingdom, 11 Manchester Academy of Health Sciences Centre, University of Manchester, Manchester, United Kingdom, 12 Devices for Dignity, National Institute of Health Research MedTech & In-vitro Diagnostics Co-operative, United Kingdom, 13 Department of Renal Medicine, Manchester University NHS Foundation Trust, Manchester, United Kingdom

* t.bampouras@lancaster.ac.uk

**Data Availability Statement:** The anonymised data set files are available from Dryad (DOI: 10.5061/dryad.3ffbg79hp).

## Abstract

### Background

Frailty is associated with adverse health outcomes in people with chronic kidney disease (CKD). Evidence supporting targeted interventions is needed. This pilot randomised controlled trial (RCT) aimed to inform the design of a definitive RCT evaluating the effectiveness of a home-based exercise intervention for pre-frail and frail older adults with CKD.

### Methods

Participants were recruited from nephrology outpatient clinics to this two-arm parallel group mixed-methods pilot RCT. Inclusion criteria were: ≥65 years old; CKD G3b-5; and Clinical Frailty Scale score ≥4. Participants categorised as pre-frail or frail using the Frailty Phenotype were randomised to a 12-week progressive multi-component home-based exercise programme or usual care. Primary outcome measures included eligibility, recruitment, adherence, outcome measure completion and participant attrition rate. Semi-structured interviews were conducted with participants to explore trial and intervention acceptability.

### Results

Six hundred and sixty-five patients had an eligibility assessment with 217 (33%; 95% CI 29, 36) eligible. Thirty-five (16%; 95% CI 12, 22) participants were recruited. Six were

**Funding:** The EX-FRAIL CKD Trial is supported by a grant from Kidney Research UK (IN_013_20180306), which has funded equipment costs and HJG's study-related activity. The Department of Renal Medicine at LTHTR provided funding for participant travel costs. ACN and HJG receive non-financial support from the National Institute for Health Research (NIHR) Lancashire Clinical Research Facility. HMLY is supported by a grant from the NIHR (DRF-2016-09-015). SM's research is supported by the NIHR Devices for Dignity Med tech Co-operative (D4D). The views expressed in this publication are those of the authors and not necessarily those of the NHS, the NIHR or the Department of Health and Social Care. The funders had no role in the study design; collection, analysis, and interpretation of the data; writing the report; and the decision to submit the report for publication.

**Competing interests:** Unrelated to this body of work, APD has received lecture fees from speaking at the invitation of MSD and received travel support from Pharmacosmos. This does not alter our adherence to PLOS ONE policies on sharing data and materials.

**Abbreviations:** ANCOVA, Analysis of Covariance; CI, Confidence Intervals; CKD, Chronic Kidney Disease; FES-I, Falls Efficacy Scale-International; FP, Frailty Phenotype; HRQOL, Health-Related Quality of Life; LTHTR, Lancashire Teaching Hospitals NHS Foundation Trust; MRC, Medical Research Council; NHS, National Health Service; NIHR, National Institute of Health Research; POS-S RENAL, Palliative Care Outcome Scale-Symptoms RENAL; RCT, Randomised Controlled Trial; RPE, Rating of Perceived Exertion; SF-12, Short Form-12v2; SPPB, Short Physical Performance Battery; TIDieR, Template for Intervention Description and Replication.

categorised as robust and withdrawn prior to randomisation. Fifteen participants were randomised to exercise and 14 to usual care. Eleven (73%; 95% CI 45, 91) participants completed ≥2 exercise sessions/week. Retained participants completed all outcome measures (n = 21; 100%; 95% CI 81, 100). Eight (28%; 95% CI 13, 47) participants were withdrawn. Fifteen participated in interviews. Decision to participate/withdraw was influenced by perceived risk of exercise worsening symptoms. Participant perceived benefits included improved fitness, balance, strength, well-being, energy levels and confidence.

## Conclusions

This pilot RCT demonstrates that progression to definitive RCT is possible provided recruitment and retention challenges are addressed. It has also provided preliminary evidence that home-based exercise may be beneficial for people living with frailty and CKD.

## Trial registration

ISRCTN87708989; https://clinicaltrials.gov/.

## Introduction

Frailty is a state of increased vulnerability to a disproportionate decline in health status when exposed to an insult, such as infection or trauma [1]. Frailty is highly prevalent in chronic conditions, including chronic kidney disease (CKD) [2], within which underlying pathological processes contribute to the development and advancement of the frailty syndrome [3]. Importantly, frailty and its precursor pre-frailty are associated with adverse health outcomes [4]. Within CKD populations, frailty is associated with worse health-related quality (HRQOL) [5], falls [6], hospitalisation [7] and mortality [8]. Validated frailty screening and assessment tools are available [9] and are increasingly used in clinical practice. However, evidence supporting targeted interventions for people living with pre-frailty/frailty is still needed [10].

Physical inactivity and associated poor physical function are common in CKD and worsen with disease progression [11,12]. Both are components of physical frailty [13,14] and are individually associated with adverse health outcomes in CKD, including mortality [15]. Ramer et al. [14] found that maintaining independence was an essential health outcome priority for older people living with CKD. Maintenance of or increasing to a regular frequency of physical activity can lead to improvement in physical frailty in older adults [16]. Greenwood et al. [17,18] demonstrated that a renal rehabilitation programme can improve physical function and is also associated with longer event-free survival in patients with CKD. Increasing physical activity levels, therefore, may lead to improved physical function and, in turn, maintenance of independence and improved survival rates in people living with frailty and CKD. However, additional high quality trials are needed [19] that include people living with frailty, individuals that are often poorly represented in interventional studies [20].

Home-based exercise programmes may be more effective in people living with frailty, as they allow practice in a familiar functionally-relevant environment. Furthermore, the benefits may be sustained in the longer term, as they are implemented without the need for direct supervision, empowering patients to incorporate exercise within their daily lives. A recent systematic review and meta-analysis of exercise, including home-based exercise, in people with non-dialysis chronic kidney disease demonstrated improvements in physical and walking

capacity [21]. However, the authors acknowledged that the generalisability of findings is "limited by age" as the "approximate mean age of participates in the included trials ranged from 50 to 65 years" [21]. Studies in older frail non-CKD populations suggest that home-based exercise interventions are feasible and may be associated with improved outcomes, in terms of frailty, functional performance, nutritional status and falls incidence [22]. However, research is needed to evaluate home-based exercise interventions tailored to the needs of older people living with frailty and CKD [23].

The aim of the EX-FRAIL CKD Trial, a pilot randomised controlled trial (RCT), is to inform the design of a definitive RCT that evaluates the effectiveness of a home-based exercise intervention in pre-frail and frail older adults with CKD by: (1) evaluating the rate of eligibility, recruitment, intervention adherence, outcome measure completion and attrition; (2) qualitatively exploring the acceptability of the randomisation procedure, outcome measures and, in the intervention arm, of the progressive home-based exercise programme; and (3) estimate the standard deviation of walking speed in pre-frail/frail people with CKD to inform the sample size calculation for a definitive RCT. A pilot RCT was needed to address key uncertainties prior to definitive evaluation of the intervention to maximise the success of a large-scale RCT.

## Methods

This section will present an abridged version of the methods; a full description of the methods has been published elsewhere [24]. Ethical approval was granted by the North West Greater Manchester East Research Ethics Committee (reference 18/NW/0211) and the National Health Service (NHS) Health Research Authority (project reference 244772).

### Trial design

The EX-FRAIL CKD trial is a two-arm parallel group pilot RCT. Participants were allocated in a ratio of 1:1 to 12-weeks of home-based exercise or usual care. Outcome assessments were performed at baseline and 12-weeks' post-randomisation. Final assessments were delayed for some participants (specifically those who temporarily had the intervention held due to an adverse event) to allow participants the opportunity to complete a total of 12-weeks of home-based exercise. Participants were invited to participate in a nested qualitative study following 12-week assessments or following a participant's decision to discontinue the exercise programme.

### Participants

Participants were recruited from Department of Renal Medicine outpatient clinics at the Lancashire Teaching Hospitals NHS Foundation Trust (LTHTR). Inclusion criteria were: age ≥65 years old; CKD G3b-5 (not receiving dialysis or received a kidney transplant); and with a Clinical Frailty Scale score ≥4 [25]. The Clinical Frailty Scale is a simple screening measure that has been validated in people with advanced CKD at risk of frailty [9,25]. Exclusion criteria were: unstable angina or recent (within the last 3 months) myocardial infarction; uncontrolled arrhythmias; persistent uncontrolled hypertension (systolic blood pressure >180 mmHg or diastolic blood pressure >110 mmHg); recent (within the last 3 months) stroke or transient ischaemic attack; registered blind; unable to mobilise independently; receiving palliative care for advanced terminal cancer; recently (within the last 12 months) enrolled in a structured exercise programme (e.g. cardiac rehabilitation) prescribed by a health professional; anticipated to commence dialysis or receive a renal transplant within the next 3 months; insufficient understanding of the English language to complete study questionnaires or follow advice within the exercise programme guidebook; and clinical and/or research team consider

participation in the exercise programme unsafe. Following written informed consent, participants underwent an objective frailty assessment, using the Frailty Phenotype (FP), to ensure that only patients with pre-frailty or frailty were randomised.

## Intervention

Table 1 describes the intervention using the Template for Intervention Description and Replication (TIDieR) checklist [26]. Briefly, exercise group participants received a physiotherapist-led exercise education session, an exercise guidebook and weekly telephone-calls from the research team. The multi-component exercise programme comprised a combination of strength, aerobic and balance exercises [24]. There were six exercises within the programme, with each having four different levels of difficulty. Participants categorised as frail were advised to perform level one exercises initially, whereas pre-frail participants could start with level two exercises. Exercise progression was discussed during weekly telephone calls with aim of maintaining a rating of perceived exertion (RPE) score of 12–16 (i.e. moderate intensity) for exercises 2–6 [27].

## Primary outcome measures

Primary outcome measures included eligibility, recruitment, intervention adherence, outcome measure completion and participant attrition rate. Reasons for ineligibility and non-consent were recorded. In the intervention arm, reasons for non-adherence were documented. Reasons for failure to complete outcome measures and for study withdrawal were also recorded.

**Table 1. TIDieR checklist.**

| Item | |
|---|---|
| **Brief name** | The EX-FRAIL CKD Exercise Programme. |
| **Rationale** | Exercise training is associated with improved health outcomes in adults with CKD. Evidence also suggests that home-based exercise interventions may improve outcomes in older adults. |
| **Materials** | Exercise guidebook, exercise diary and wrist/ankle weights. |
| **Procedures** | Exercise education session and weekly telephone-calls. |
| **Provider** | Exercise education was delivered by a physiotherapist. Telephone calls were performed by a physiotherapist or specialist trainee with relevant experience. |
| **Modes of delivery** | Face-to-face exercise education session followed by weekly telephone calls. |
| **Location** | Exercise education sessions were delivered in a private room at NIHR Lancashire Clinical Research Facility. Exercises were completed in a participant's own home. |
| **Frequency and duration** | Participants received an education session that lasted approximately 60 minutes. Participants aimed to perform 3 exercise sessions at home per week, each lasting approximately 30–45 minutes. |
| **Tailoring** | Initial exercise levels were determined by frailty status, unless the physiotherapist determined otherwise due to safety concerns. If a participant could perform any of the exercises comfortably after week 1, exercise progression was discussed with the participant. |
| **Modifications** | An alternative exercise was provided if a participant was unable to perform an exercise as originally intended. If a participant was unable to complete the proposed repetitions, they were advised to perform a lower number initially. |
| **Adherence and fidelity** | Exercises were delivered as described in the exercise guidebook. If modification was needed, the participant was provided additional documentation. Adherence was assessed during telephone calls and through review of exercise diaries. Outcomes of telephone calls were discussed within the research team. See also 'Intervention Adherence' in 'Results' section. |

CKD, chronic kidney disease. NIHR, National Institute of Health Research.

Progression criteria are recommended for pilot trials to assess whether there should be progression to a definitive RCT [28]. Trial progression criteria were determined *a priori* by the research team: (1) eligibility: stop <5%, go >10%; (2) recruitment: stop <10%, go >30%; (3) exercise adherence: stop: <30%, go >70%; (4) outcome measure completion: stop <70%, go >80%; and (5) loss to follow-up: stop >50%, go <25%.

## Secondary outcome measures

An overview of secondary outcome measures is provided below; a detailed description has been published previously [24]. All measures were performed at baseline and at 12-week follow-up visits.

1. **Physical Function**: Physical function was assessed by measuring walking speed and the Short Physical Performance Battery (SPPB) [29].

2. **Frailty**: Frailty was assessed using the original FP [13]. Participants were categorised as frail if 3 or more FP components were present and as pre-frail if 1 or 2 FP components were present.

3. **Activities of Daily Living**: The Barthel Index questionnaire was used to evaluate independence with 10 activities of daily [30].

4. **Falls**: The Falls Efficacy Scale-International (FES-I) questionnaire was used to assess fall concern [31]. The number of falls within the preceding 6 months was also recorded.

5. **Symptom-Burden**: The Palliative Care Outcome Scale-Symptoms RENAL (POS-S RENAL) questionnaire was used to assess symptom burden [32].

6. **HRQOL**: The Short Form-12v2 (SF-12) was used to assess HRQOL and was used to generate physical and mental health summary measures (PCS and MCS, respectively) [33].

## Interviews

Semi-structured interviews were conducted, using a predetermined topic guide, with a purposively selected group of participants from both study arms, considering age, sex and frailty status. Interviews explored the acceptability of the randomisation procedure, outcome measures and, in the intervention arm, of the progressive home-based exercise programme. The Chief Investigator (ACN), a Specialist Trainee in Renal Medicine, conducted all interviews. ACN received training from an experienced qualitative researcher (KWF). During interviews ACN was cognisant of the potential for personal bias, given his prior association with participants and the subject area. Interviews and transcripts were regularly discussed and reviewed with KWF for potential bias concerns. Further detail on the interviews has been published previously [24].

## Sample size

The target sample size was 40 participants, which allowed for a dropout rate of up to 50% and would still provide sufficient data to assess study feasibility and inform a sample size calculation for a definitive RCT [34,35]. A sample of 12–14 participants was the goal for the qualitative study and anticipated to achieve data saturation.

## Randomisation and blinding

A central, concealed web-based randomisation process (www.sealedenvelope.com) was performed in blocks of 4 with stratification limited to one factor, FP status. Blinding of

participants was not possible due to the nature of the intervention. Blinding of outcome assessors was not performed in this pilot RCT for pragmatic reasons.

## Data analyses

Quantitative outcome measures are reported descriptively with 95% confidence intervals (CI). Analysis of Covariance (ANCOVA) was performed on normally distributed data relating to secondary outcome variables to descriptively present the mean difference (and associated standard deviation) between groups whilst adjusting for baseline measurements. Barthel Index score data were not normally distributed and are therefore presented as median and interquartile range (IQR). Data relating to frailty status, a categorical variable, are presented as frequencies and percentages. SPSS Statistics (version 25.0.0.1, IBM Corp) and R (version 4.0.2, R Foundation for Statistical Computing) statistical software were used to conduct statistical analyses. The intervention effect size (Cohen's d) was calculated and used to inform the sample size calculations for a future trial. G*Power (version 3.1.9.4) was used to perform the sample size calculations [36]. Qualitative data was analysed using thematic analysis; narrative segments were coded and then collated into potential themes, which were iteratively reviewed to develop a 'thematic map' [37]. NVivo software (version 12.6.0, QSR International) was used to support qualitative analysis. Qualitative and quantitative findings were linked [38] and presented in a 'joint display' [39].

## Results

### Participant recruitment

The trial opened in August 2018; data collection was completed in December 2019. Fig 1 demonstrates the participant flow throughout the study. Six hundred and sixty-five patients had an eligibility assessment with 217 (33%; 95% CI 29, 36) eligible for enrolment. Four hundred and forty-eight (67%; 95% CI 64, 71) patients were considered ineligible. Reasons for ineligibility are detailed in S1 Table. One hundred and fifty-three (23%; 95% CI 20, 26) patients declined participation. Most patients did not offer a reason for declining participation (n = 79; 52%; 95% CI 43, 60). Reported reasons for declining participation are detailed in S2 Table. The research team were unable to contact 28 (4%; 95% CI 3, 6) patients to complete the eligibility assessment and were unable to accommodate a baseline study visit for 1 (0.46%; 95% CI 0.02, 2.94) eligible patient prior to recruitment closure. A total of 35 (16%; 95% CI 12, 22) participants were recruited to the study. Six (17%; 95% CI 7, 34) were categorised as robust using the FP assessment and were withdrawn prior to randomisation. Fifteen participants were randomised to the exercise group and 14 to the usual care group.

### Participant demographics and clinical characteristics

Table 2 details participant demographics and clinical characteristics. Four (29%; 95% CI 10, 58) participants were categorised as frail in the usual care group compared to 5 (33%; 95% CI 13, 61) in the exercise group. The remaining participants were categorised as pre-frail.

### Progression criteria results

A median of 28 (IQR 16) exercise sessions were completed during the 12-week intervention period. Eleven (73%; 95% CI 45, 91) exercise group participants completed ≥2 exercise sessions per week, with a mean of 36.5±8.5 minutes spent exercising each session. The mean RPE score for exercises 2–6 was 12±2. The main reasons for missing exercise sessions were pain

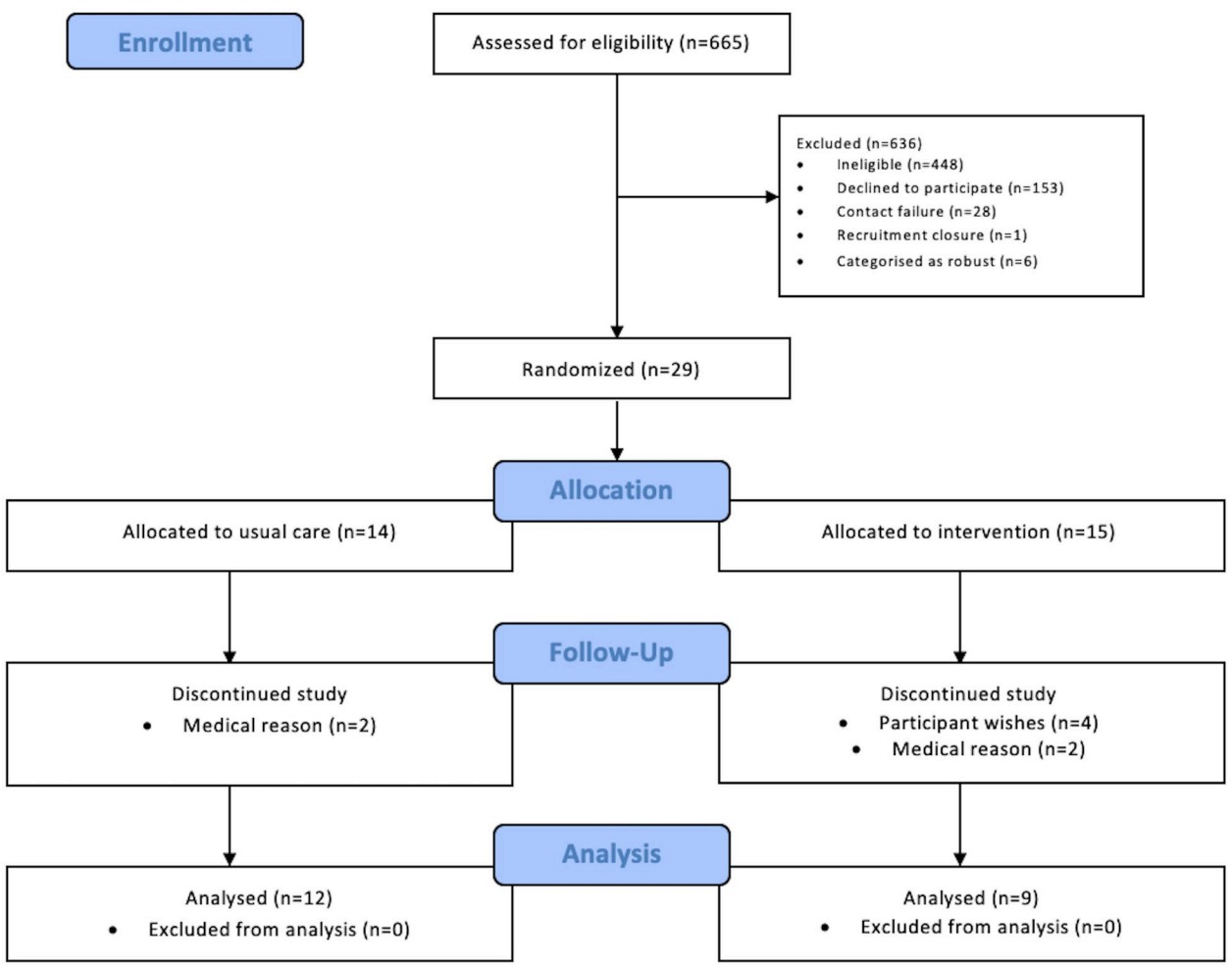

**Fig 1. Study flow diagram.**

(n = 14; 56%; 95% CI 35, 75), participant wishes (n = 7; 28%; 95% CI 13, 50) and feeling unwell (n = 4; 16%; 95% CI 5, 37).

Retained usual care and exercise group participants completed all outcome measures (n = 21; 100%; 95% CI 81, 100). Eight (28%; 95% CI 13, 47) participants were withdrawn from the study. Two (14%; 95% CI 3, 44) participants were withdrawn from the usual care group; the research team learned issues following randomisation that meant participation in an exercise programme was considered unsafe. Six (40%; 95% CI 17, 67) participants were withdrawn from the exercise group: musculoskeletal pain (n = 3; all made a complete recovery), prescribed specific exercise by a physiotherapist during the trial period (n = 1) and participant wishes (n = 2).

### Physical function and patient reported outcome measures

Table 3 presents between-group differences in physical function and patient reported outcomes. The adjusted mean group difference in walking speed and SPPB between exercise and usual care groups were: 0.01 metres/second (95% CI -0.07, 0.10) and 0.5 (95% CI -0.9, 1.8), respectively. The adjusted mean group difference in FESI, POS-S RENAL, SF-12 PCS and SF-12 MCS were: 3.4 (95% CI -3.5, 10.3), -1.4 (95% CI -6.6, 3.7), -3.9 (95% CI -9.3, 1.5) and 0.2

**Table 2. Participant demographics and clinical characteristics.**

| | Usual Care (n = 14) | Exercise (n = 15) |
|---|---|---|
| Age, years, mean ±SD | 78.8 ±7.0 | 77.0 ±8.3 |
| Female, n (%) | 7 (50) | 6 (40) |
| Ethnicity | | |
| • White British, n (%) | 14 (100) | 15 (100) |
| Primary Renal Disease, n (%) | | |
| • Renovascular/Ischaemic | 7 (50) | 5 (33) |
| • Diabetic | 2 (14) | 0 (0) |
| • Cardio-renal | 0 (0) | 2 (13) |
| • Glomerulonephritis | 1 (7) | 3 (20) |
| • Obstructive Uropathy | 0 (0) | 2 (13) |
| • Other | 3 (21) | 3 (20) |
| • Unknown | 1 (7) | 0 (0) |
| CCI, mean ±SD | 3.4 ±0.9 | 4.2 ±1.3 |
| Medications, median (IQR) | 8.5 (6.3) | 9.0 (5.0) |
| Smoking History, n (%) | | |
| • Non-smoker | 4 (29) | 5 (33) |
| • Ex-smoker | 8 (57) | 8 (53) |
| • Current smoker | 2 (14) | 2 (13) |
| Living Circumstances, n (%) | | |
| • Alone | 8 (57) | 8 (53) |
| • With Family | 6 (43) | 7 (47) |
| Received Carer Support, n (%) | 0 (0) | 2 (13) |
| CFS Score, n (%) | | |
| • 4: Vulnerable | 8 (57) | 10 (67) |
| • 5: Mildly frail | 2 (14) | 5 (33) |
| • 6: Moderately frail | 4 (29) | 0 (0) |
| Frailty Phenotype | | |
| • Pre-Frail | 10 (71) | 10 (67) |
| • Frail | 4 (29) | 5 (33) |
| Fall Previous 6 Months, n (%) | 1 (7) | 0 (0) |
| BMI, kg/m$^2$, mean ±SD | 29.4 ±5.7 | 29.4 ±6.9 |
| Blood Pressure, mean ±SD | | |
| • Systolic, mmHg | 142.6 ±11.5 | 139.5 ±18.5 |
| • Diastolic, mmHg | 70.1 ±10.6 | 70.4 ±9.4 |
| Laboratory Variables, mean ±SD | | |
| • Creatinine, μmol/L | 239.5 ±63.4 | 274.4 ±106.1 |
| eGFR, ml/min/1.73m$^2$ | 20.4 ±7.2 | 18.9 ±7.0 |
| • Haemoglobin, g/L | 117.6 ±15.6 | 117.1 ±6.7 |
| • Albumin, g/L | 42.9 ±3.9 | 40.5 ±2.5 |

CCI, Charlson Comorbidity Index, BMI, Body Mass Index; eGFR, estimated Glomerular Filtration Rate.

Data presented as number (%), mean ± SD or median (IQR).

**Table 3. Between-group differences in physical function and patient reported outcomes.**

| Outcome Measure | Usual Care Group (n = 12) Mean ±SD (95% CI) | | | Exercise Group (n = 9) Mean ±SD (95% CI) | | | Unadjusted between-group differences Mean (95% CI) | Adjusted between-group differences Mean (95% CI) |
|---|---|---|---|---|---|---|---|---|
| | Baseline | Follow-up | Change | Baseline | Follow-up | Change | | |
| Walking Speed (m/s) | 0.73±0.18 (0.61, 0.84) | 0.72±0.22 (0.58, 0.86) | -0.01±0.09 (-0.07, 0.05) | 0.77±0.29 (0.55, 0.99) | 0.77±0.24 (0.58, 0.95) | 0.003 (-0.068, 0.074) | 0.05 (-0.16, 0.26) | 0.01 (-0.07, 0.10) |
| SPPB | 7.7±2.6 (6.0, 9.3) | 7.8±2.7 (6.0, 9.5) | 0.1±1.5 (-0.9, 1.0) | 7.8±2.0 (6.2, 9.4) | 8.3±2.0 (6.8, 9.9) | 0.6±1.4 (-0.5, 1.7) | 0.6 (-1.7, 2.8) | 0.5 (-0.9, 1.8) |
| FESI | 32.2±7.1 (27.6, 36.7) | 28.8±8.6 (23.4, 34.3) | -3.3±6.4 (-7.4, 0.7) | 32.6±13.5 (22.2, 42.9) | 32.6±14.1 (21.7, 43.4) | 0.0±8.8 (-6.7, 6.7) | 3.7 (-6.6, 14.1) | 3.4 (-3.5, 10.3) |
| POS-S Renal | 12.2±6.0 (8.3, 16.0) | 13.3±6.6 (9.2, 17.5) | 1.2±6.3 (-2.8, 5.1) | 15.1±11.5 (6.3, 24.0) | 13.9±9.6 (6.5, 21.2) | -1.2±5.7 (-5.6, 3.2) | 0.6 (-6.8, 7.9) | -1.4 (-6.6, 3.7) |
| SF-12 PCS | 35.8±7.7 (30.9, 40.6) | 38.9±4.4 (36.1, 41.7) | 3.2±6.7 (-1.1, 7.4) | 39.1±9.0 (32.2, 46.0) | 36.8±9.4 (29.5, 44.0) | -2.3±7.0 (-7.7, 3.1) | -2.2 (-8.6, 4.3) | -3.9 (-9.3, 1.5) |
| SF-12 MCS | 48.1±11.6 (40.8, 55.5) | 49.9±9.4 (43.9, 55.9) | 1.8±8.1 (-3.4, 6.9) | 51.0±8.3 (44.7, 57.4) | 51.9±8.8 (45.1, 58.6) | 0.8±7.3 (-4.8, 6.4) | 2.0 (-6.5, 10.4) | 0.2 (-6.2, 6.6) |

m/s, metres/second; SPPB, Short Physical Performance Battery; FESI, Falls Efficacy Scale-International; POS-S RENAL, Palliative care Outcome Scale-Symptoms RENAL;

SF-12 PCS, Short Form-12v2 Physical Component Summary; SF-12 MCS, Short Form-12v2 Mental Component Summary.

Data presented for participants that completed follow-up assessments (n = 21).

(95% CI -6.2, 6.6), respectively. The median Barthel Index scores for the usual care group at baseline and follow-up were 95 (IQR 5; 95% CI 95, 100) and 95 (IQR 9; 95% CI 90, 100), respectively. The median Barthel Index scores for the exercise group at baseline and follow-up were 100 (IQR 8; 95% CI 90, 100) and 100 (IQR 8; 95% CI 90, 100), respectively. Table 4 presents the frailty status change for usual care and exercise groups. The relative risk for improvement in frailty status with exercise was 4.0 (95% CI 0.7, 25.6).

## Adverse events

There were 32 adverse events in the exercise group and 22 in the usual care group. There were 12 adverse reactions (i.e. adverse events related to the intervention): musculoskeletal pain (9), fall (1), nocturnal leg cramps (1) and postural dizziness (1). There were no adverse events related to the trial outcome measures. Within the exercise group, there were 2 serious adverse events (hospitalisations due to an infection) unrelated to the intervention.

## Sample size estimation

The calculated mean change (and associated standard deviation) in outcome measures presented in Table 3 were used for sample size estimation. With an alpha of 0.05%, calculations indicated that to achieve 80% power a minimum sample size of 1542, 268 and 200 participants

**Table 4. Frailty status change for usual care and exercise groups.**

| Frailty Status Change | Usual Care | | Exercise | |
|---|---|---|---|---|
| | Frequency | Percentage (95% CI) | Frequency | Percentage (95% CI) |
| Improved | 1 | 8 (0.2, 38.5) | 3 | 33 (7.5, 70.1) |
| Unchanged | 9 | 75 (42.8, 94.5) | 5 | 56 (21.2, 86.3) |
| Worse | 2 | 17 (2.1, 48.4) | 1 | 11 (0.3, 48.3) |

**Table 5. Progression criteria: Joint display of quantitative and qualitative results.**

| | Thresholds | Total | Percentage (95% CI) | Qualitative Results | Inferences |
|---|---|---|---|---|---|
| **Eligibility** | **STOP**: <5% **GO**: >10% | 217 | 33 (29, 36) | No discussion. | Silence |
| **Recruitment** | **STOP**: <10% **GO**: >30% | 35 | 16 (12, 22) | Factors affecting decision to participate included altruism, potential for personal gain and influence of family/health professionals. Mixed feelings about randomisation process. Some concerns about ability to participate in exercise due to own mobility issues or because of risk of pain. | Complementary |
| **Exercise adherence** | **STOP**: <30% **GO**: >70% | 11 | 73 (45, 91) | Exercise adherence influenced by staff attitude/support, participant personality traits and personal goals, participant fear of pain/injury, exercise difficulty and participant perceived benefit/lack of benefit. | Complementary |
| **Outcome measure completion** | **STOP**: <70% **GO**: >80% | 21 | 100 (81, 100) | Highlighted importance of participants understanding/accepting outcome assessments. | Complementary |
| **Lost to follow-up (including withdrawn)** | **STOP**: >50% **GO**: <25% | 8 | 28 (13, 47) | Participant decision to withdraw influenced by perceived exercise ineffectiveness, discomfort experienced during exercise and a fear of future injury | Complementary |

would be needed to determine differences between usual care and exercise group participants' walking speeds, SPPB scores and POS-S Renal scores, respectively, following the exercise intervention.

## Qualitative results

Fifteen participants agreed to take part in interviews. Seven participants had been randomised to usual care and 8 participants to exercise. One exercise participant withdrew from the study prior to completion of the exercise intervention but agreed to participate in the qualitative study. Qualitative study participant demographics are presented in S3 Table. Identified themes were related to the feasibility of the trial and the intervention itself. S4 Table presents these themes alongside supportive quotes. A joint display of the quantitative and qualitative results with regards to the study progression criteria is presented in Table 5.

Many participants' decision to take part in the study was motivated by a sense of altruism, whereas others were motivated by the potential for personal gain. Participants' decision-making was also influenced by family members and trusted healthcare professionals. Although many were indifferent about the randomisation process, some had definite preferences and were disappointed with the outcome of randomisation. Some participants acknowledged that they felt frail and were not deterred by the word 'frail'. One participant did not identify as being frail by their understanding of the term. Overall, participants understood the rationale for the outcome assessments and accepted their inclusion within the study visits. One participant reported frustration with the patient-reported outcome measures (PROMs), specifically with regards to the number of questions asked and the time taken to complete the PROMs. They did not highlight any one PROM being more problematic than another.

Some participants expressed fear, particularly fear of pain, at the thought of participating in exercise or being more physically active. Several participants highlighted the importance of positive staff attitude when delivering exercise education and supporting participant engagement with exercise. Participants found the concept of RPE unfamiliar. Only one participant remained dismissive of recording RPE scores. A variety of factors influenced participants' motivation to exercise including: personal goals; self-determination and resilience; personal responsibility to participate after study consent; a sense of achievement following exercise session completion; telephone calls with the research team; exercise location; exercise enjoyment

(or lack thereof); and perceived ineffectiveness of exercise. Participants reported liking exercising at home, citing convenience, flexibility and privacy as being positive aspects of the exercise programme. Several participants described the benefits that they experienced, including: improved fitness, balance and strength. Participants also described improvements in their well-being, energy levels and confidence. Finally, involvement in the study promoted self-reflection, in terms of personal levels of physical activity and functional ability, which motivated participants to be more active.

## Discussion

To our knowledge, this is the first trial involving a home-based exercise programme for pre-frail and frail older adults with CKD. Progression criteria thresholds were exceeded for eligibility, adherence and outcome measure completion. However, recruitment and loss to follow-up progression criteria thresholds were not achieved. The use of a mixed-methods approach provided a comprehensive evaluation of study procedures and the intervention, highlighting potential areas requiring adaptation.

Although, the use of the word 'frail' did not deter participants from taking part in our study, the term can be viewed negatively [40]. The inclusion of this in the study materials may have deterred some patients from discussing study involvement. Rather than avoid this terminology, study materials could provide a more detailed explanation of the language of frailty. Participants described concerns about their ability to participate in exercise due to perceived mobility issues or because of concerns about exercise eliciting or exacerbating pain. These findings highlight the importance of early face-to-face discussions between potential participants and informed healthcare professionals to address these concerns. There were also mixed feelings about the randomisation process with some participants having a preferred study arm. Adoption of a delayed-start trial design would hopefully dispel these reservations, though would necessitate a longer study period [41].

There is an anticipated attrition rate with any study, not least studies involving an older, frail and multimorbid population [42]. Studies of home exercise interventions for older adults living with frailty have reported retention rates ranging from 53–98% [22]. In our study, decision to withdraw appeared to be influenced by perceived ineffectiveness and a fear of worsening symptom experience with physical activity. These findings have been reported previously in patients with CKD [43,44]. A previous study that evaluated 12 weeks of supervised exercise demonstrated an improvement in symptom-burden for people with CKD [45]. There was an increase in the frequency of 'joint/bone pain' with the intervention that included resistance training; however, this was not statistically significant [45]. Importantly, the intervention resulted in a reduction of 'loss of muscular strength/power' symptoms [45] and an objective improvement in muscle mass and strength [46]. Further education about the potential benefits of exercise on symptom experience should be provided to participants from the outset, whilst acknowledging that it is not uncommon to experience transient discomfort when increasing physical activity levels.

With a home-based exercise intervention, participants require the appropriate knowledge, skills and confidence to actively engage with the intervention, otherwise known as patient activation [47]. Low patient activation levels are described in older patients living with frailty [48] and older patients living with advanced CKD [49]. Additional efforts to improve patient activation, specifically tailored to the individual participant [50], may promote participant retention. Considering the theory of planned behaviour, positive enhancement of an individual's perceived behavioural control, i.e. the "perception of the ease or difficulty of performing the behaviour of interest", is beneficial [51]. Furthermore, self-determination theory suggests

intrinsic motivation is an important factor for behaviour change; therefore encouraging participant autonomy may improve participant retention [52].

Although confidence intervals were expectedly wide, exercise group participants had a 0.5 (95% CI -0.9, -1.8) score increase in SPPB- an acknowledged meaningful score change [50]. Notably, there was also an improvement in POS-S RENAL score (-1.4; 95% CI -6.6, 3.7), though again confidence intervals were wide. Participants reported experiencing benefits related to exercise, including improved fitness, balance, strength, well-being, energy levels and confidence. Although this study was not powered to investigate the effectiveness of the home-based exercise programme, these findings suggests that the intervention may offer benefits to people living with frailty and CKD.

A sample size of 1542 participants would be needed for a definitive RCT that used walking speed as the primary outcome measure. Even before accounting for participant attrition, this is clearly an unrealistic target. A more realistic sample size of 268 and 200 participants would be needed if using the SPPB or POS-S Renal, respectively. Physical function and symptom-burden are clinical outcomes relevant and important for people living with CKD and people living with frailty alike [14,15,53–55]. Therefore, we suggest that either may be used as the primary outcome measure of interest in a RCT investigating the effectiveness of a home-based exercise programme in this patient population.

Notwithstanding this pilot RCT's strengths, there are acknowledged limitations. Recruited participants were all White British and the feasibility of a RCT in other populations cannot be presumed. For pragmatic reasons, patients that declined enrolment in the main study were not offered the opportunity to participate in the qualitative study. Therefore, reasons for non-enrolment can only be reported descriptively based upon patient comments. Most participants enrolled in the study were categorised as pre-frail by the FP. It is possible that patients living with frailty, particularly more severe frailty, have greater concerns about participating in exercise, either due to lack of confidence or burden of frailty and co-morbidity. There is therefore a risk that the suggested trial adaptations do not adequately address this potential recruitment issue. Exercise adherence was measured using exercise diaries and weekly telephone calls, which are subject to recall bias. However, this was considered the most realistic approach as using an accelerometer would: (1) not distinguish between exercise and other activity; and (2) introduce an additional parameter that may influence behaviour change, i.e. participants using the accelerometer may be more motivated to exercise. Participants involved in the qualitative study were interviewed by a researcher also involved in the delivery of the intervention. However, participants were informed that the purpose of the interview was to understand their experience of the study to identify areas for improvement for a definitive RCT. Finally, blinding of outcome assessors should be considered for a RCT investigating the effectiveness of the intervention.

In summary, the EX-FRAIL CKD trial demonstrates that progression to a large-scale definitive RCT is possible provided recruitment and retention challenges are addressed. Furthermore, it has determined the necessary sample size for a RCT using clinically relevant and important primary outcome measures for the study population. Finally, it has provided preliminary evidence that home-based exercise may be beneficial for people living with frailty and CKD.

## Supporting information

**S1 Table. Reasons for ineligibility.**
(DOCX)

**S2 Table. Reported reasons for declining study participation.**
(DOCX)

**S3 Table. Qualitative study: Participant demographics and clinical characteristics.**
(DOCX)

**S4 Table. Qualitative study: Themes and supporting quotes.**
(DOCX)

**S1 File. CONSORT checklist.**
(DOC)

**S2 File. First Ethics approved protocol.**
(PDF)

## Acknowledgments

We would like to thank Claire Corless, Maya Leach, Helen Cross, Samantha McCulloch, Alexandra McCarrick and Wanda Ingham for supporting study procedures, including recruitment activity, participant visits and data collection. We would also like to thank the LTHTR Lay Research Group and patient representatives that were members of the EX-FRAIL CKD Trial Steering Committee.

## Author Contributions

**Conceptualization:** Andrew C. Nixon, Theodoros M. Bampouras, Helen J. Gooch, Hannah M. L. Young, Kenneth W. Finlayson, Neil Pendleton, Sandip Mitra, Mark E. Brady, Ajay P. Dhaygude.

**Data curation:** Andrew C. Nixon.

**Formal analysis:** Andrew C. Nixon, Kenneth W. Finlayson.

**Funding acquisition:** Andrew C. Nixon, Theodoros M. Bampouras, Helen J. Gooch, Hannah M. L. Young, Kenneth W. Finlayson, Neil Pendleton, Sandip Mitra, Mark E. Brady, Ajay P. Dhaygude.

**Investigation:** Andrew C. Nixon, Theodoros M. Bampouras, Helen J. Gooch.

**Methodology:** Andrew C. Nixon, Theodoros M. Bampouras, Helen J. Gooch, Hannah M. L. Young, Kenneth W. Finlayson, Neil Pendleton, Sandip Mitra, Ajay P. Dhaygude.

**Project administration:** Andrew C. Nixon.

**Supervision:** Sandip Mitra, Mark E. Brady, Ajay P. Dhaygude.

**Writing – original draft:** Andrew C. Nixon.

**Writing – review & editing:** Andrew C. Nixon, Theodoros M. Bampouras, Helen J. Gooch, Hannah M. L. Young, Kenneth W. Finlayson, Neil Pendleton, Sandip Mitra, Mark E. Brady, Ajay P. Dhaygude.

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
