## [Decision Letter · Decision Letter 0]

13 Jan 2021

PONE-D-20-35665

Home-based exercise for people living with frailty and chronic kidney disease: a mixed-methods pilot randomised controlled trial

PLOS ONE

Dear Dr. Nixon,

Thank you for submitting your manuscript to PLOS ONE. Both reviewers agree that it provides a clear and well presented account of your trial, but they highlight a few minor points of concern.  After careful consideration, therefore, we feel that it has merit but does not fully meet PLOS ONE’s publication criteria as it currently stands. We invite you to submit a revised version of the manuscript that addresses the points raised during the review process.

We look forward to receiving your revised manuscript.

Kind regards,

Antony Bayer

Academic Editor

PLOS ONE

Journal Requirements:

2.) In your Methods section, please provide additional information about the eligibility criteria and about the intervention.

3.) When reporting the results of qualitative research, we suggest consulting the COREQ guidelines: http://intqhc.oxfordjournals.org/content/19/6/349. In this case, please consider including more information on the number of interviewers, their training and characteristics; how participants were selected; and whether bias issues were considered.

4.) Thank you for stating the following in the Competing Interests section:

'I have read the journal's policy and the authors of this manuscript have the following competing interests: Unrelated to this body of work, APD has received lecture fees from speaking at the invitation of MSD and received travel support from Pharmacosmos.'

5.) We noted in your submission details that a portion of your manuscript may have been presented or published elsewhere: 'An abstract was presented at ERA-EDTA that detailed preliminary results: https://doi.org/10.1093/ndt/gfaa142.P0958'.

Please clarify whether this publication was peer-reviewed and formally published. If this work was previously peer-reviewed and published, in the cover letter please provide the reason that this work does not constitute dual publication and should be included in the current manuscript.

6.) We note that you have indicated that data from this study are available upon request. PLOS only allows data to be available upon request if there are legal or ethical restrictions on sharing data publicly. For information on unacceptable data access restrictions, please see http://journals.plos.org/plosone/s/data-availability#loc-unacceptable-data-access-restrictions.

Reviewers' comments:

Reviewer's Responses to Questions

**Comments to the Author**

1. Is the manuscript technically sound, and do the data support the conclusions?

Reviewer #1: Partly

Reviewer #2: Yes

2. Has the statistical analysis been performed appropriately and rigorously? 

Reviewer #1: Yes

Reviewer #2: I Don't Know

3. Have the authors made all data underlying the findings in their manuscript fully available?

Reviewer #1: No

Reviewer #2: Yes

4. Is the manuscript presented in an intelligible fashion and written in standard English?

Reviewer #1: Yes

Reviewer #2: Yes

5. Review Comments to the Author

Reviewer #1: A pilot randomized controlled trial was conducted to inform the design of a definitive RCT evaluating the effectiveness of home-based exercise intervention in older adults with chronic kidney disease. Primary outcome measures included eligibility, recruitment, adherence, outcome measure completion, and attrition rate. Fifteen participants were randomized to intervention and 14 to usual care. Participant perceived benefits in the areas of fitness, balance, strength, well-being, energy levels and confidence. The study identified challenges with recruitment and retention.

Minor revisions:

1- Line 247: Indicate the statistical method(s) used to estimate the 95% confidence intervals.

2- Line 344: The power calculations should include: (1) the estimated outcomes in each group; (2) the α (type I) error level; (3) the statistical power (or the β (type II) error level); (4) the target sample size and (5) for continuous outcomes, the standard deviation of the measurements.

Reviewer #2: Nixon et al. reported a pilot RCT regarding home-based exercise for frail elderly patients with advanced CKD. The study is very well structured and performed, and the manuscript is well written. Also, it is a very important and relevant area of research to explore the efficacy and effectiveness of very practical and applicable way of long-term exercise such as home-based exercise in those with CKD. However, current version of manuscript is still immature to be published as it appears.

* Although the outcomes are various, several studies of home-based exercise in CKD were already reported elsewhere (Eidemak et al. Nephron 1997, Tang et al. Int J Nurs Pract. 2017, Hiraki et al. Clin Exp Nephrol 2017, Aoike et al. Clin Exp Nephrol 2018, all of which were cited in recent metanalysis paper ; Nakamura et al. Sci Rep 2020) and all of those should be cited and compared to current study in terms of methods/durations/intensities of the exercise, adherence to and completion rate of the protocols, and the outcomes.

* One of the major limitations of this study is that exercise was only measured subjectively (diaries and phone calls). This should be also noted and discussed as a major limitation of the study.

* Intensity of the exercise seems relatively weak for those with almost normal Barthel Index. Authors should also discuss about this, which might affected the judgement of effect of the exercise.

* In Table 3, all the data are presented as mean +- SD, however, I wonder if all data were distributed normally or not.

6. PLOS authors have the option to publish the peer review history of their article (what does this mean?). If published, this will include your full peer review and any attached files.

Reviewer #1: No

Reviewer #2: No

---

## [Author Response · Author response to Decision Letter 0]

15 Apr 2021

Dear Editorial Board,

We thank the Editors and Reviewers for taking the time to consider our manuscript ‘Home-based exercise for people living with frailty and chronic kidney disease: a mixed-methods pilot randomised controlled trial’ for publication as a clinical trial in PLOS One. Thank you also for the extension to the time usually permitted for a response. We endeavour to respond to the Academic Editor’s and Reviewers’ points comprehensively below.

Academic Editor

1. We have now formatted the manuscript to meet PLOS ONE’s style requirements.

2. We have provided additional information about the eligibility criteria and about the intervention.

3. We have provided additional information on qualitative methodology.

4. We have updated our Competing Interests section.

5. An abstract of preliminary results was presented at an ERA-EDTA conference and the abstract was published in a NDT supplementary issue, associated with the ERA-EDTA conference. The manuscript under consideration for publication in PLOS ONE, however, contains the full details of the study and updated results that have not been published previously. 

6. The minimal anonymized data set necessary to replicate study findings is now available at https://doi.org/10.5061/dryad.3ffbg79hp.

Reviewer 1

1. We are uncertain as to what is meant by the statistical methods used to calculate the 95% confidence intervals. All 95% confidence intervals were calculated using the described statistical software, e.g. 95% confidence intervals of proportions using R (version 4.0.2, R Foundation for Statistical Computing). We would be happy to revise again if the Reviewer could provide further detail on this point.

2. Thank you for this comment. We have updated the section titled ‘Sample Size Estimation’ to improve clarity. 

Reviewer 2

1. Thank you for highlighting these papers. We have amended the Introduction to acknowledge the systematic review and meta-analysis, which includes all the mentioned papers. 

2. Thank you for highlighting this. We acknowledge that exercise diaries and telephone calls are subject to recall bias, though do not entirely agree that the use of exercise diaries and telephone calls is a major limitation of the study. We have amended the limitations section of the Discussion to address this point. 

3. Although participants had a high Barthel Index score, they were assessed as pre-frail or frail using the Frailty Phenotype. There is overlap between disability and frailty, but an individual can be frail without having disability; frailty itself can affect ability to participate in physical activity. Moreover, The American College of Sports Medicine and American Heart Association Guidelines highlight the importance of physical activity being increased gradually in older adults, particularly those that are very deconditioned. Participants used the Borg Rating of Perceived Exertion (RPE) to report their perceived exertion during each exercise session; these scores were used to guide the progression to more challenging exercise levels each week. The intended RPE exercises 2-6 was achieved (mean RPE score for exercises 2-6 was 12+/-2). Considering the above, we do not believe that the intensity of the exercise programme affected the trial feasibility outcomes.

4. Barthel Index scores are now presented as median and interquartile range. The results are presented in the Results section titled ‘Physical Function and Patient Reported Outcome Measures’.

Thank you for considering our revised manuscript. We hope that you are satisfied with our responses and we look forward to hearing from you.

Yours sincerely,

Dr Andrew Nixon 

Renal Medicine and General Internal Medicine Specialist Trainee

---

## [Editor Report · Decision Letter 1]

30 Apr 2021

Home-based exercise for people living with frailty and chronic kidney disease: a mixed-methods pilot randomised controlled trial

PONE-D-20-35665R1

Dear Dr. Nixon,

Thank you for your thoughtful responses to the issues raised by your original submission. We’re pleased to inform you that your revised manuscript has been judged scientifically suitable for publication and will be formally accepted for publication once it meets all outstanding technical requirements.

Kind regards,

Antony Bayer

Academic Editor

PLOS ONE
---

## [Editor Report · Acceptance letter]

22 Jun 2021

PONE-D-20-35665R1 

Home-based exercise for people living with frailty and chronic kidney disease: a mixed-methods pilot randomised controlled trial 

Dear Dr. Bampouras:

I'm pleased to inform you that your manuscript has been deemed suitable for publication in PLOS ONE. Congratulations! Your manuscript is now with our production department. 

Kind regards, 

on behalf of

Professor Antony Bayer 

Academic Editor

PLOS ONE